# Port Distributed Energy Management Considering USVs Charging and Discharging in Polymorphic Network

Qi Qu
*Navigation College*
*Dalian Maritime University*
Dalian, China
15164007491@163.com

Fei Teng*
*College of Marine Electrical Engineering*
*Dalian Maritime University*
Dalian, China
brenda teng@163.com

Qi Xu
*Research Institute of Intelligent Networks*
*Zhejiang Lab*
Hangzhou, China
xuqi@zhejianglab.com

*Abstract*—To reduce the environmental impact of port carbon emissions and promote the sustainable development of ports, this paper proposes a port distributed energy management strategy considering the charging and discharging of unmanned surface vehicles (USVs) under the polymorphic network. Firstly, taking into account the trend of continuous automation of ports, data centers are used to meet the ports' growing computing power needs. A data center power consumption calculation model considering data processing delay constraints is proposed. Secondly, taking the port microgrid operating cost, the main grid electricity purchase or sales cost, carbon cost, data center operating cost and USVs charging and discharging cost as objective functions, a port distributed energy management model considering USVs charging and discharging is established. Then, a distributed algorithm based on mixed integer linear programming is proposed to solve the problem. Finally, simulation results demonstrate the effectiveness of the proposed method.

*Keywords—port microgrid, data center, distributed energy management*

## I. INTRODUCTION

With the increasing severity of global warming, the development and use of renewable energy is increasingly recognised as a key solution in the world [1]. As important hubs for global trade and logistics [2], ports ' energy consumption and carbon emissions have a significant impact on climate change. Therefore, it is very important to optimize the energy management system of the port to reduce the carbon footprint of the port and develop the green port.

Port energy management is the process of effectively managing and utilizing the port's energy resources and is an optimization problem [3]. Iris [4] proposed a integer model approach to address the issue of uncertain power generation in port. On the basis of solving the problem of uncertain power generation of renewable energy power generation, Kermani [5] coordinate and optimize all equipment energy consumption, such as cranes, refrigerated boxes and offices to improve port operations and energy performance. With the development of information technology, ports are increasingly inclined towards automation and digital transformation. The data center, as an efficient resource utilization optimization equipment, has the capability of analysis and processing of data, and can satisfy the requirement of computing capacity of the port [6]. Much research has been done on the power consumption of the data center. Yu [7] minimized the cost of data centers by minimizing the cost of energy and carbon emissions as an objective function. Yu [8] has used a low carbon economy to simulate the uncertainty of power prices, renewables, and data center loads in order to minimise the long term running costs of a data center microgrid. Therefore, how to safely achieve port energy management needs to be studied.

Because of the distribution features of the equipment and the load in a large scale harbor, it is possible to use a distributed algorithm to resolve the port energy management. Huang [9] proposed a no-delay distribution algorithm to minimize the overall operational cost. Chen [10] proposed an economical scheduling algorithm based on distributed, fast and economical scheduling to ensure the economy of the port. Although there have been studies on distributed algorithms for port energy management, considering the charging and discharging of USVs is a mixed integer linear programming problem, as the addition of integer variables makes the solution space of the problem discontinuous and exponentially growing. Especially for large-scale problems, preprocessing of the problem is required, using parallel computing or distributed computing techniques, which is different from previous research. Furthermore, in the process of integration of the port microgrid and computing power network, it is necessary to communicate information among different devices in the port. However, the traditional IP network structure is not able to satisfy the communication demand of the port equipment. Therefore, this paper proposes a port energy management system considering the charging and discharging of USVs under the polymorphic network.

To sum up, the innovative points of this paper are as follows:

- To lower the operational cost of the port microgrids, the cost of buying and selling power from the main grid, the carbon price, the operation cost of the data centre and the USVs, a port power management system is built which takes into account the charging and discharging of USVs in the polymorphic network. The system can ensure communication between heterogeneous distributed devices.

This work was supported by the National Natural Science Foundation of China (Grant No. 52201407, 61751202), and the Zhejiang Lab Open Research Project (Grant No. K2022QA0AB03). (Corresponding author: Fei Teng).

- A distributed energy management method is proposed based on a mixed integer linear programming algorithm to achieve energy management of the port.

## II. PORT ENERGY MANAGEMENT SYSTEM UNDER POLYMORPHIC NETWORK

### A. Port structure

The distribution of energy in the port depends on the exchange of information among different devices. The demand for real-time, reliable and secure is very high. The conventional IP network structure can not satisfy this demand. Polymorphic networks support more efficient communication protocols and wider communication ranges, enabling USVs to interact with charging facilities more stably and quickly. USVs can obtain real-time information such as the status and location of charging facilities, to develop more reasonable charging plans. In this paper, a port energy management system which takes into account USAs' charging and discharging is presented, which includes data level, control level and service level. The data layer is used to store the related data of the port energy management. The control level is responsible for domain management and authorization management. The service level is in charge of the implementation of the port distributed energy management. Define different routes according to business needs to achieve management between polymorphic networks and ensure distributed energy management in ports, as shown in Fig. 1.

On the basis of these hypotheses, the objective function of the computing model of the port data center is expressed as

$$P_{IDC,n}^{t} = \alpha\, m_{n}^{t} + \beta$$
$$\alpha = PUE * P_{n,\min}$$
$$\beta = \left[ PUE * L^{t} \left( P_{n,\max} - P_{n,\min} \right) \right] \Big/ \mu_{n}$$

(1)

where $P_{IDC,n}^{t}$ is the power consumption of the port data center, $m_{n}^{t}$ is the number of active servers in the data center, $\alpha, \beta$ are the energy consumption coefficient of the data center, and The Power Usage Effectiveness (PUE) is the proportion between the overall energy use of the data centre and the energy consumption for the calculation and storage, which is set to 1.2. $P_{n,\max}, P_{n,\min}$ are the maximum and silent capacity of one server in a data center, which are set to 2 kWh and 1 kWh. $L^{t}$ is the data load generated by the port, $\mu_{n}$ is the active server's service speed, which is set to 40 number/h. $T = \{1, 2, \cdots, t\}$ is the number of time slots.

To make sure that a port data centre operates reliably, the restrictions are

- **Server quantity constraints** Port data center has a limitation on the number of servers, which can be represented by

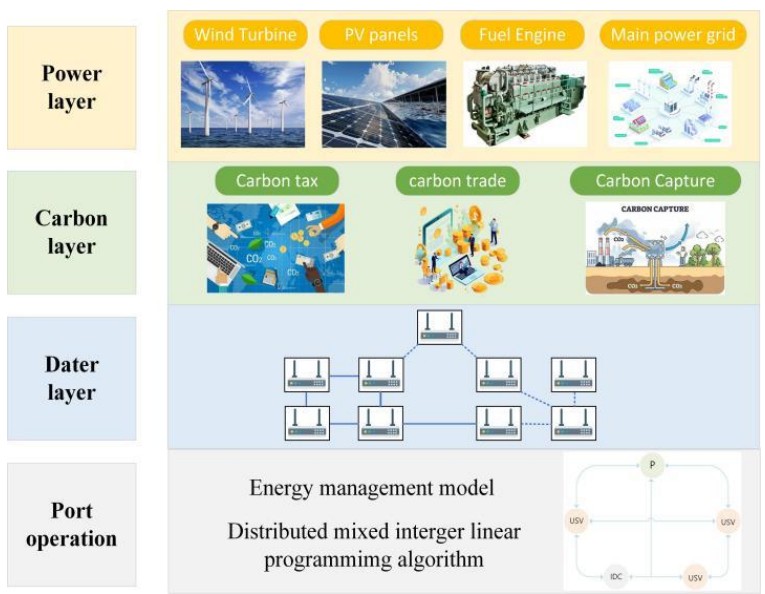

Fig. 1. Energy management systems for ports.

### B. Port data center power consumption calculation model

In order to build the computing model for data center, it is necessary to take into account the extra influence caused by the devices and the load of the port. The hypothesis is as follows: (1) The data load is a delayed sensitivity load. (2) Processing time of data delay is no more than one hour. (3) Port data center servers are all uniform [11].

$$0 \le m_{n}^{t} \le M_{n}$$

(2)

where $M_{n}$ is the total number of servers in the data center, which is set to 600 number.

- **Data load delay constraints** To guarantee the continuous operation of the port, we adopt the M/M/1

queue model to calculate the mean stay time to keep it within the latency.

$$0 < \frac{1}{\mu_n - \frac{L^t}{m_n^t}} \leq Q \qquad (3)$$

where $Q$ is the maximum latency time of the port data load, which is set to 0.5 s.

## III. Port energy management method under polymorphic network

### A. Port energy management model under polymorphic network

To reduce the cost of port operations, this paper proposes to consider introducing a carbon tax to encourage ports to reduce carbon emissions by increasing the cost of fossil energy. Therefore, the objective function includes five parts, namely, port operation cost, main grid cost, carbon tax cost, USVs charging and discharging cost, and data center operation cost.

$$F = \min\{F_1 + F_2 + F_3 + F_4 + F_5\} \qquad (4)$$

where $F$ is the total cost, $F_1$ is the port operating cost, $F_2$ is the main power grid cost, $F_3$ is the carbon tax cost, $F_4$ is the unmanned ship charging and discharging cost, and $F_5$ is the data center operating cost.

The port operation cost is composed of photovoltaic energy supply cost, wind energy supply cost and fuel energy supply cost, which is expressed as follows

$$\begin{aligned} F_1 &= F_w + F_{pv} + F_{fu} \\ &= \sum_{n_d=1}^{V_d} \alpha_w P_{n,w} + \sum_{n_d=1}^{V_d} \alpha_{pv} P_{n,pv} + \sum_{n_d=1}^{V_d} \alpha_{fu} P_{n,fu} \end{aligned} \qquad (5)$$

where $F_w, F_{pv}, F_{fu}$ are the costs of wind power generation, photovoltaic power generation and fossil fuel power generation, $P_{n,w}, P_{n,pv}, P_{n,fu}$ are the output power of wind turbines, PV panels and fuel engines, and $\alpha_w, \alpha_{pv}, \alpha_{fu}$ are the cost coefficients of power generation devices.

The cost function of the main power grid buying electricity is expressed as follows

$$F_2 = K_M P_M \qquad (6)$$

where $K_M$ is the cost coefficient of the main grid purchasing electricity. $P_M$ is the power supply of the main grid.

Using carbon tax to manage carbon emissions in ports is as follows

$$F_3 = K_C \sum_{n_d=1}^{V_d} \iota P_{n,fu} \qquad (7)$$

where $K_C$ is the carbon tax price of 50\$ [12] and $\iota$ is the coefficient of carbon dioxide produced by fossil fuel combustion, which is set to 0.01 .

The USVs charging and discharging cost is expressed as follows

$$F_4 = \sum_{n_s=1}^{V_s} P_S \left( K_{SA} a_n - K_{SB} b_n \right) \qquad (8)$$

where $P_S$ is the charging and discharging power, $K_{SA}, K_{SB}$ are the charging and discharging cost coefficients. $a_n, b_n$ are the charging and discharging states.

Data centre operation cost is as follows

$$F_5 = K_D \sum_{n_D=1}^{V_D} P_{IDC,n}^t \qquad (9)$$

where $K_D$ is the data center power consumption cost coefficient, which is set to 0.670 \$/kWh.

To guarantee that the port operates reliably, it is necessary to establish the following restrictions.

- **Power balance restriction** To guarantee the proper running of a harbour, it is necessary to satisfy the demand for electricity in the harbour.

$$\sum_{n_d=1}^{V_d} P_{n,w} + \sum_{n_d=1}^{V_d} P_{n,pv} + \sum_{n_d=1}^{V_d} P_{n,fu} + P_M \geq P_{load} + \sum_{n_D=1}^{V_D} P_{IDC,n}^t \qquad (10)$$

where $P_{load}$ is the load demand of the port.

- **Power generation devices capacity restrictions** To guarantee the safety of the electric power system, it is necessary to control the electricity production in a certain scope. The restrictions are as follows

$$\begin{aligned} P_w^{\min} &\leq P_{n,w} \leq P_w^{\max} \\ P_{fu}^{\min} &\leq P_{n,fu} \leq P_{fu}^{\max} \\ P_{pv}^{\min} &\leq P_{n,pv} \leq P_{pv}^{\max} \end{aligned} \qquad (11)$$

where $P_w^{\min}, P_w^{\max}, P_{fu}^{\min}, P_{fu}^{\max}, P_{pv}^{\min}, P_{pv}^{\max}$ are the upper and lower limits of the power of each power generation device.

- **USVs charging and discharging restrictions** The operating constraints of USVs can be described as

$$e_n^0 = \mathrm{E}_n^{init}$$

$$e_n = e_n^0 + P_S\left(\theta_n^A a_n - \theta_n^B b_n\right)$$

$$e_n \geq \mathrm{E}_n^{ref} \tag{12}$$

$$\mathrm{E}_n^{min} \leq e_n \leq \mathrm{E}_n^{max}$$

$$a_n + b_n \leq 1$$

$$P^{min} \leq \sum_{n_s=1}^{V_s} P_S\left(a_n - b_n\right) \leq P^{max}$$

where $e_n$ is the USVs energy level, $\mathrm{E}_n^{init}, \mathrm{E}_n^{ref}, \mathrm{E}_n^{min}, \mathrm{E}_n^{max}$ are the initial energy storage, calculated reference energy storage, minimum energy storage and maximum energy storage, respectively, and $\theta_n^A, \theta_n^B$ are the energy level coefficients.

- **Data center operation restrictions** To meet the port's computing power requirements, the operation of the data center must comply with the following constraints

$$0 \leq m_n^t \leq M_n$$

$$0 < \frac{1}{\mu_n - \dfrac{L^t}{m_n^t}} \leq Q \tag{13}$$

In summary, the port energy management model can be summarized as follows

$$\min \sum_{i=1}^{V} F_i\left(P_{n,i}\right)$$

$$st: \begin{cases} \sum_{n_d=1}^{V_d} P_{n,w} + \sum_{n_d=1}^{V_d} P_{n,pv} + \sum_{n_d=1}^{V_d} P_{n,fu} + P_M \geq P_{load} + \sum_{n_D=1}^{V_D} P_{IDC,n}^t \\ P^{min} \leq \sum_{n_s=1}^{V_s} P_S\left(a_n - b_n\right) \leq P^{max} \\ \Theta = \Theta_d \bigcup \Theta_{ship} \bigcup \Theta_{IDC} \end{cases}$$

$$\begin{cases} \Theta_{n,d} = \left\{ \begin{bmatrix} P_{n,w} \\ P_{n,pv} \\ P_{n,fu} \\ P_M \end{bmatrix} \in R^{VT}\left(n=1,\cdots,V\right) \right\} \\ \Theta_{n,ship} = \left\{ \begin{bmatrix} a_n \\ b_n \\ e_n \\ 0 \end{bmatrix} \in R^{VT}\left(n=1,\cdots,V\right) \right\} \\ \Theta_{n,IDC} = \left\{ \begin{bmatrix} m_n^t \\ 0 \\ 0 \\ 0 \end{bmatrix} \in R^{VT}\left(n=1,\cdots,V\right) \right\} \end{cases} \tag{14}$$

where $V_d, V_s, V_D$ are the number of power nodes, USVs nodes and data center nodes, respectively. $\Theta_d, \Theta_{ship}, \Theta_{IDC}$ are the set of power nodes, USVs nodes and data center nodes, respectively.

### B. Distributed Mixed Integer Linear Programming Algorithm

There are $V$ devices in the port energy system, and the communication network topology between them can be represented by a directed graph $G = \{W, E_h\}$, where $W = \{1,2,3\cdots,V\}$ is the node set and $E_h = \{(j,i): \phi_j^i(h) > 0\}$ is the set of directed edges. In this directed graph, if there exist $\eta \in (0,1)$, $i,j \in \{1,\ldots,V\}$, $h > 0$, such that $\phi_j^i(h) \in [0,1), \phi_i^i(h) \geq \eta$ and $\phi_j^i(h) > 0$, then there is $\phi_j^i(h) \geq \eta$. Furthermore, for $h \geq 0$, when $i = \{1,\ldots,V\}$ occurs, there is $\sum_{j=1}^{V} \phi_j^i(h) = 1$; when $j = \{1,\ldots,V\}$ occurs, there is $\sum_{i=1}^{V} \phi_j^i(h) = 1$. All device nodes in the port energy system can generate information interaction and have strong connectivity. For $i$ each $(j,i) \in E_\infty$, there is a positive integer $T \geq 1$, and agent receives information from the adjacent agent $j$ at least once every $T$ consecutive iterations.

Before designing the algorithm, it is necessary to establish some initial assumptions.

**Assumption 1.** In the presence of $i = \{1,\ldots,V\}$, functions $y_i^T Z_i : R^{VT} \to R$, $\Lambda_i \subseteq R^{VT}$ and $A_i Z_i - d$ are all convex and set $\Lambda_i$ is a compact subset of $R^{VT}$.

**Assumption 2.** There exists $\tilde{Z} = \left[\tilde{Z}_1 \ldots \tilde{Z}_V\right]^\top \in \mathrm{relint}(\Lambda)$, where $\mathrm{relint}(\Lambda)$ is a relative interior of the set $\Lambda$, in $Z$ such that the part $\sum_{i=1}^{V} A_i \tilde{Z}_i - d$ in $\sum_{i=1}^{V} A_i \tilde{Z}_i - d \leq 0$ is linear and the other parts $\sum_{i=1}^{V} A_i \tilde{Z}_i - d < 0$.

**Assumption 3.** When $\{y(h)\}_{h \geq 0}$ is a monotonically decreasing sequence of positive real numbers, and $y(h) \leq y(r)$ is satisfied for all $h \geq g \geq 0$, there are: (1) $\sum_{h=0}^{\infty} y(h) = \infty$; (2) $\sum_{h=0}^{\infty} y(h)^2 < \infty$, $y(h) = \vartheta/(h+1)$, $\vartheta > 0$.

When Assumption 1-3 are true, there is strong duality and an optimal primal-dual pair $(Z^*, \lambda^*)$, $Z^* = \left[Z_1^* \ldots Z_V^*\right]^\top$ exists. The following formula can be obtained.

$$L(Z^*, \lambda) \leq L(Z^*, \lambda^*) \leq L(Z, \lambda^*), \lambda \in R_+^{VT}, Z \in \Theta \tag{15}$$

| Algorithm Distributed Mixed Integer Linear Programming Algorithm |
| --- |
| 1:Initialization: $h=0$, $\hat{Z}_i(0)\in\Theta_i, \lambda_i(0)\in R_+^{4T}, i=1,\cdots,V$. |
| 2:Repeat |
| 3:$h\leftarrow h+1$. |
| 4:Update $q_i(h)$ based on $q_i(h)=\sum_{j=1}^{V}\phi_j^i(h)\lambda_j(h)$. |
| 5:Update $Z_i(h+1)$ based on $$Z_i(h+1)\in\arg\min_Z\left(y_i^T Z_i(h)+q_i^T(h)A_iZ_i(h)-l_i^T(h)\frac{m}{V}\right).$$ |
| 6:Update $\lambda_i(h+1)$ based on $$\lambda_i(h+1)=\left[0,q_i(h)+y_hA_iZ_i(h+1)-y_h\frac{m}{V}\right]^+.$$ |
| 7:Update $\hat{Z}_i(h+1)$ based on $$\hat{Z}_i(h+1)=\frac{\left[\sum_{g=0}^{h-1}y_gZ_{ir}\right]+y_hZ_i(h+1)}{y_h+\left[\sum_{g=0}^{h-1}y_g\right]}.$$ |
| 8. Until convergence |

It can be found from (14) that the port's energy management is a mixed integer linear programming problem.

$$\min_{\{Z_i\in Z_i\}_{i=1}^V}\sum_{i\in V}c_i^TZ_i$$
$$st:\sum_{i=1}^{V}A_iZ_i-d\leq 0 \tag{16}$$

where $V$ is the number of nodes and $Z_i$ is the decision variable.

The Lagrangian function is expressed as

$$L(Z,\lambda)=\sum_{i=1}^{V}L_i(Z_i,\lambda)=\sum_{i=1}^{V}\left(y_i^TZ_i+\lambda^T(A_iZ_i-d)\right) \tag{17}$$

where $Z=\left[Z_1^\top\cdots Z_V^\top\right]^\top\in\Theta\subseteq R^n, n=\sum_{i=1}^{V}VT,\lambda\in R_+^{4T}$ are vectors of Lagrange multipliers.

Thus the dual function can be expressed, and the objective function. And The object and constraint function in the dual problem are separable, so it can be expressed as

$$f(\lambda)=\sum_{i=1}^{V}f_i(\lambda)=\sum_{i=1}^{V}\min_{Z_i\in\Theta_i}L_i(Z_i,\lambda) \tag{18}$$

where, $f(\lambda)$ is expressed as the dual function of node $i$.

The dual problem is thus as follows

$$\mathcal{O}:\max_{\lambda\geq 0}\sum_{i=1}^{V}f_i(\lambda) \tag{19}$$

Here, the solution of the dual question (19) corresponds to the solution of the initial question (16). Based on the above, the paper presents a distribution algorithm which is based on the dual decomposition of the mixed integer linear programming [13] to address this issue.

In addition, because of the bad estimation of the Lagrange multiplier at the beginning of the algorithm, it will affect the performance of the algorithm. Therefore, this paper uses two different sequences $\hat{Z}_i(h),\tilde{Z}_i(h)$ to correct the result $Z_i(h)$. $Z_i(h)$ is the weighted average and optimal solution of the port energy management problem, which is expressed as follows:

$$\tilde{Z}_i(h+1)=\begin{cases}\hat{Z}_i(h+1) & h<h_{s,i}\\\dfrac{\sum_{g=h_{s,i}}^{h}y(g)Z_i(g+1)}{\sum_{g=h_{s,i}}^{h}y(g)} & h\geq h_{s,i}\end{cases}$$

## IV. SIMULATION

In this section, this paper applies Matlab to validate the validity of the distributed algorithm. The port consists of 1 wind turbine, 1 photovoltaic panel, 1 fuel engine device, 1 data center and 3 USVs. These are classified as power, USVs, and data Nodes, as illustrated in Fig. 2. For the electric generating apparatus, the cost coefficients are [0.610, 0.615, 0.650, 0.600]. The top and bottom limits of the generating apparatus are [1100, 1100, 900], [0,0,200]. Assume that between 10 and 11 a.m., the port operating load is 1900 kWh and the current port data load generation rate is 10,000/h. The results of the simulation are displayed in Fig. 3 to 5.

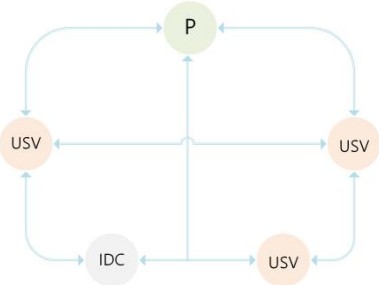

Fig. 2. Node classification.

Fig. 3 illustrates the output power of every power generating unit in the port energy management system and the power consumption of the port data center. The power supplied by each power generation equipment is [707.38, 652.24, 609.25, 579.11], the total power supply of the port is 2547.98 kWh. The electricity cost is 1576.11$, and the carbon tax generated is 304.625$. From the figure, it can be seen that the

burning of fossil fuels will pollute the environment and generate a lot of CO2, and bring additional costs, the output power of fuel generators is relatively low, and ports are more likely to choose to use clean energy to operate. The number of servers turned on by the data center to handle the data load arriving at this moment is 264, the power consumption is 616.80kWh, and the cost is 413.26$.

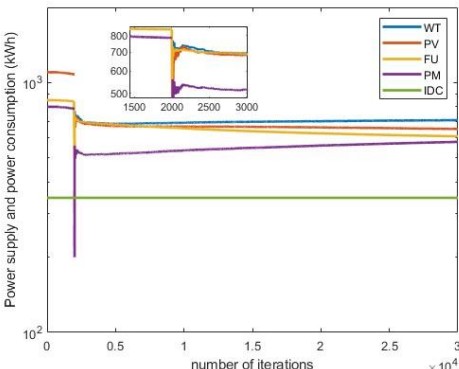

Fig. 3. Power supply of power supply devices and power consumption of data center.

The charging and discharging status of USVs is shown in Fig. 4. All three USVs are charged, and the battery energy has increased significantly, which are [5.86, 5.26, 6.75], and the cost is 17.10 $. The total cost of port operation is 2311.095 $. In this paper, taking into account USVs' charging and discharging in polymorphic networks, this paper presents a mixed integer linear programming distributed algorithm with globally coupled constraints. The algorithm iteration process involves dual variables with different global coupling constraints. After 30,000 iterations of simulation, the results are shown in Fig. 5. When the number of iterations reaches 20,000, all dual variables converge to the same value, and the dual variable values corresponding to the optimal solution to the port energy management problem are obtained, which further proves the effectiveness of the algorithm proposed in this paper.

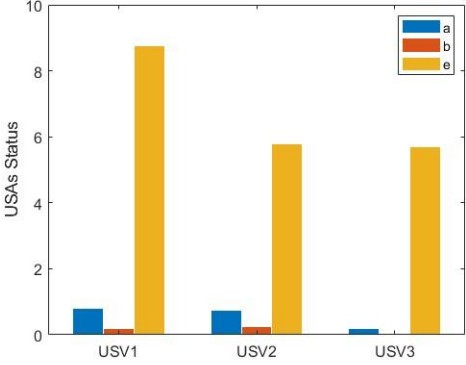

Fig. 4. USAs battery energy and charging and discharging state

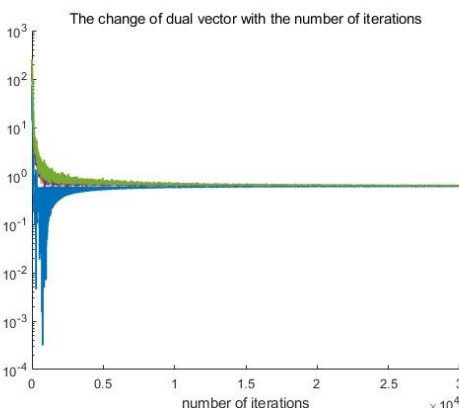

Fig. 5. Dual variable iteration process.

## V. Conclusion

This paper proposes a port distributed energy management method considering the charging and discharging of USVs under a polymorphic network, and establishes a data center power consumption calculation model to meet the port computing resource requirements. And taking the port microgrid operating cost, the main grid electricity purchase or sales cost, carbon cost, data center operating cost and USVs charging and discharging cost as objective functions, a port distributed energy management model considering USVs charging and discharging is constructed. A distributed algorithm based on mixed integer linear programming is used to solve the problem, and the experimental results show that the proposed approach is effective.

In this paper, only one data centre is used to satisfy the requirement of computation. Along with the rapid progress in the field of port automation, it may be possible to combine several data centres to achieve the service of a port to make sure that the port operates efficiently.

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
