# OpenReview forum: "Port Distributed Energy Management Considering USVs Charging and Discharging in Polymorphic Network"
_IEEE.org/ICIST/2024/Conference — IEEE ICIST 2024 Conference Submission_

### Official Review · Reviewer_N8RN · 2024-08-30
**This paper can be accepted**

**Rating:** 9
**Confidence:** 4

**Review:**

1. Could the authors provide more detail on what is meant by a "polymorphic network" and how it specifically impacts the charging and discharging processes of the unmanned surface vehicles (USVs)?
2. The use of a mixed integer linear programming (MILP) algorithm is noted, but could the authors describe any specific challenges encountered with this approach, and how they were addressed?

---

### Official Review · Reviewer_NjBe · 2024-09-02
**This paper can be accepted.**

**Rating:** 7
**Confidence:** 5

**Review:**

This paper proposes a distributed port energy management of unmanned surface vehicles under the polymorphic network.
The reviewer's comments are as follows:
1.The font in Figure 1 should be larger for better readability.
2.The English grammar and format of this manuscript could be further polished and checked carefully.
3.The format of references is not uniform.
4.The effectiveness of the method can be more clearly reflected by properly enlarging the simulation result graph

---

### Decision · Program_Chairs · 2024-09-06

Accept (Oral)